# Circadian Pattern of Acute Myocardial Infarction and Atrial Fibrillation in a Mediterranean Country: A study in Diabetic Patients

**DOI:** 10.3390/medicina57010041

**Published:** 2021-01-06

**Authors:** Stylianos Daios, Christos Savopoulos, Ilias Kanellos, Christos Argyrios Goudis, Ifigeneia Nakou, Stergiani Petalloti, Nicolas Hadjidimitriou, Dimitrios Pilalas, Antonios Ziakas, Georgia Kaiafa

**Affiliations:** 1First Propedeutic Department of Internal Medicine, Department of Medicine, Aristotle University of Thessaloniki, AHEPA University Hospital, Stilponos Kyriakidi Street, 54636 Thessaloniki, Greece; stylianoschrys.daios@gmail.com (S.D.); eliotkan@yahoo.com (I.K.); petalste@ser.forthnet.gr (S.P.); hn@freemail.gr (N.H.); pilalas_jim@hotmail.com (D.P.); gdkaiafa@yahoo.gr (G.K.); 2Cardiology Department, Serres General Hospital, 62120 Serres, Greece; cgoudis@hotmail.com (C.A.G.); ifigenianakou@gmail.com (I.N.); 3First Department of Cardiology, Aristotle University of Thessaloniki, AHEPA University Hospital, 54636 Thessaloniki, Greece; tonyziakas@hotmail.com

**Keywords:** circadian rhythm, acute myocardial infarction, atrial fibrillation, diabetes mellitus, Mediterranean country

## Abstract

*Background and objectives:* The circadian pattern seems to play a crucial role in cardiovascular events and arrhythmias. Diabetes mellitus is a complex metabolic disorder associated with autonomic nervous system alterations and increased risk of microvascular and macrovascular disease. We sought to determine whether acute myocardial infarction (AMI) and atrial fibrillation (AF) follow a circadian pattern in diabetic patients in a Mediterranean country. *Materials and Methods:* This retrospective study included 178 diabetic patients (mean age: 67.7) with AMI or AF who were admitted to the coronary care unit. The circadian pattern of AMI and AF was identified in the 24-h period (divided in 3-h and 1-h intervals). Patients were also divided in 3 groups according to age; 40–65 years, 66–79 years and patients older than 80 years. A chi-square goodness-of-fit test was used for the statistical analysis. *Results:* AMI seems to occur more often in the midnight hours (21:00–23:59) (*p* < 0.001). Regarding age distribution, patients between 40 and 65 years were more likely to experience an AMI compared to other age groups (*p* < 0.001). Autonomic alterations, working habits, and social reasons might contribute to this phenomenon. AF in diabetic patients occurs more frequently at noon (12:00–14:59) (*p* = 0.019). *Conclusions:* Diabetic patients with AMI and AF seem to follow a specific circadian pattern in a Mediterranean country, with AMI occurring most often at midnight hours and AF mostly at noon. Autonomic dysfunction, glycemic fluctuations, intense anti-diabetic treatment before lunch, and patterns of insulin secretion and resistance may explain this pattern. More studies are needed to elucidate the circadian pattern of AMI and AF in diabetic patients to contribute to the development of new therapeutic approaches in this setting.

## 1. Introduction

Myocardial infarction (MI) and atrial fibrillation (AF) represent a major cause of morbidity and mortality worldwide [1,2]. Various circulatory parameters, such as heart rate (HR), blood pressure (BP), and activity of the autonomic nervous system (ANS) or renin-angiotensin cascade, demonstrate a time-of-day variation [2,3,4]. The BP circadian pattern exhibits a peak early in the morning, and dysregulation of BP is independently associated with poor prognosis in patients with acute myocardial infarction (AMI) [5]. The onset of AMI and sustained ventricular arrhythmias shows a time-of-day variation as well [6,7,8]. Circadian rhythm refers to a biological process that modulates the internal 24-h clock of the organism and regulates the cycles of alertness and sleepiness [1]. The circadian pattern has also been observed in sudden cardiac death, stroke, unstable angina, and arrhythmia [9]. Therefore, the interpretation of the link between the internal clock and cardiovascular (CVS) function may lead to the establishment of a more constructive therapeutic approach to CVS diseases.

A specific circadian periodicity has been observed in AMI in Mediterranean Caucasians, with most of the events occurring between the afternoon and midnight. Several factors may be the cause of different circadian pattern, such as cultural factors like evening-nap “siesta” and smoking [10]. Sunshine hours, increased temperature, meal distribution, and other cultural characteristics may affect the cardiac rhythm during the whole day and modify the AMI’s typical pattern [10]. More specifically, increased BP and HR was reported after the evening-nap in a similar manner after waking up in the morning [10].

Diurnal variation of the ANS activity and peripheral circadian clock may also demonstrate time-of-day variation in the electrophysiological parameters of cardiomyocytes [11,12,13]. Specifically, Brain and muscle Arnt-like protein 1 (BMAL1) binds and initiates the expression of Sodium voltage-gated channel alpha subunit 5 (Scn5a) gene in cardiomyocytes that may result in HR variability. The onset of arrhythmias may show a first peak in the morning by a following second peak in the noon [6]. However, in patients with diabetes mellitus (DM), this pattern may be altered.

Autonomic neuropathy (AN) is a serious and common complication of DM. Cardiac AN is a known complication of DM and is characterized by parasympathetic denervation, unregulated sympathetic activity, and subsequent sympathetic denervation [14]. In the clinical setting, significantly lower HR variability, a marker of autonomic dysfunction, has been identified in patients with DM [15]. Various studies demonstrated that abnormalities in the circadian rhythm of autonomic tone might lead to an altered circadian periodicity of cardiovascular events (CVE) [16]. ANS also plays an important role in the initiation and perpetuation of atrial fibrillation (AF) [17,18]. In our retrospective study, we evaluated 178 diabetic patients admitted to the coronary care unit (CCU) of a secondary hospital due to an AMI or AF. We investigated whether a circadian pattern exists regarding AMI and AF occurrence in diabetic patients in a Mediterranean country.

## 2. Methods

### 2.1. Study Design and Sample

This is a retrospective study of diabetic patients with AMI (STEMI and N-STEMI) and AF occurring in General Hospital of Serres, with the contribution of the First Propedeutic Department of Internal Medicine, AHEPA university hospital, Thessaloniki, Greece. A convenience sample of 178 patients with DM admitted to the coronary care unit (CCU) participated in the study. The criteria for the diagnosis of AMI were based on the guidelines of the European Society of Cardiology (ESC) and included criteria such as clinical symptoms consistent with myocardial ischaemia (persistent chest pain or discomfort), persistent ST-segment elevation or (presumed) a new left bundle-branch block; Electrocardiogram (ECG) changes with segment elevation ST ≥ 0.1 mv in at least two limb leads and/or ≥0.2 mv in at least two precordial leads, and elevated biochemical markers of myocardial necrosis (Creatinine Kinase- Myocardial Band—CK-MB, troponin). Criteria for the diagnosis of AF were also based on the ESC guidelines and included rhythm documentation using an electrocardiogram (ECG) showing the typical pattern of AF (absolutely irregular RR intervals and no discernible, distinct P waves). The diagnosis of DM was based on the medical records and the patient self-reported history. Patients with AMI or AF and a history of DM greater than 5 years were included in our study.

### 2.2. Data Collection

Data were collected by form completion, including demographic and clinical characteristics of patients. Data collection performed during a 6 month period (March to August 2019). The admission of the patient to the CCU was documented by the consultant cardiologist. The time of onset of AMI and AF was documented on the patient’s reports by the responsible physician. Patients were divided into 3 groups of age for calculation purposes. The first group included patients from 40 to 65 years, the second group from 66 to 79 years, and the third group older than 80. The 24-h period of the day was divided into 1- and 3-h intervals for calculation purposes. Patient cardiovascular (CV) risk factors included hypertension, dyslipidaemia, smoking, DM for more than 5 years, and family history of CV disease. Demographic characteristics included age and gender. Patients with AMI or AF who died at the emergency department or during their hospitalization were not included in the study because we could not obtain the medical records of the patients. Our hospital’s system does not allow the possibility of recording and creating a medical record for the patients who either visited the emergency department and were not hospitalized or died. All the data were collected at bedside by the consultant cardiologist. The documents required by the European and National Bioethics Committee were signed by the scientific council, CCU director, and cardiology department.

### 2.3. Statistical Analysis

A descriptive statistical analysis was performed. Continuous variables (e.g., age) were presented as means and standard deviations. Categorical variables were presented as frequencies and percentages. Patient clinical and demographic characteristics were analyzed. The day was distributed into 1- and 3-h periods. Patients were divided into 3 groups based on the age (40 to 65, 66 to 79 years, patients older than 80 years). AMI and AF were first tested for differences among the eight equal 3-h intervals and then among a 24-h distribution using the chi-square goodness-of-fit test. The three age groups were also analyzed in correlation with AMI and AF using the chi-square goodness-of-fit test. The statistical significance level was set at 0.05. Data analysis was performed using mainly the software SPSS version 22.

## 3. Results

The study population included 178 diabetic patients with AMI and AF that admitted to the CCU (Figure 1). The mean age of the patients was 67.7 years (SD = 12). Sex distribution was 123 males and 55 females. Patients’ reason for admission included AMI (*n* = 119) and AF (*n* = 59). The baseline characteristics of the patients are presented in Table 1, classified by age group in Table 2.

A significant difference (*p* < 0.001) was found when we divided the patients with AMI at 24-h equal intervals with three peaks at 22:00–22:59 (15.1%), 14:00–14:59 (10.1%), and 02:00–02:59 (8.4%). When we divided those patients at 3-h equal intervals (00:00–02:59, 03:00–05:59, 06:00–08:59, 09:00–11:59, 12:00–14:59, 15:00–17:59, 18:00–20:59, 21:00–23:59), we observed peaks at 21:00–23:59 (23.5%), 12:00–14:59 (15.1%), and 00:00–02:59 (16.8%), showing a significant circadian variation (*p* < 0.001). The event distribution is presented in Figure 2 (1-h intervals) and Figure 3 (3-h intervals). Regarding age distribution, patients between 40 and 65 years were more likely to experience an AMI than those of the other age groups (*p* < 0.001).

Regarding AF events, at 24-h distribution, a significant variation was found, with 2 peaks at 12:00–12:59 (16.9%) and 13:00–13:59 (10.2%) (*p* = 0.038). Moreover, at 3-h distribution intervals, AF demonstrated a circadian peak at 12:00–14:59 (19.7%) (*p* = 0.019). The event distribution is presented in Figure 4 (1-h intervals) and Figure 5 (3-h intervals). Additionally, the age group with the most frequent hospitalizations due to AF was the second (66 to 79 years) (*p* < 0.001). 

When we combined AMI and AF together, we found significant difference at 24-h distribution, with 4 peaks at 22:00–22:59 (11.2%), 14:00–14:59 (7.9%), 12:00–12:59 (7.9%), and 02:00–02:59 (6.7%) (*p* < 0.002). At 3-h distribution, a significant difference was found, with three-time peaks at 21:00–23:59 (19.1%), 12:00–14:59 (19.7%), and 00:00–02:59 (15.7%) (*p* < 0.001). The age distribution of both events is presented in Figure 6.

## 4. Discussion

We investigated whether there is a circadian pattern in the occurrence of AMI and AF in diabetic patients in a Mediterranean country. The circadian pattern demonstrates a significant impact on CVE and arrhythmias. Our study presents a circadian pattern of AMIs mainly at midnight hours (21:00–23:59) and a higher incidence of them in patients in 40 to 65 years. Regarding the AF initiation, the events occurred more frequently at noon (12:00–14:59). In particular, the circadian rhythm influences CVS physiology, promoting alterations in HR, BP, cardiac output, and endothelial function, and subsequently in the incidence of different CVE, such as AMI, and arrhythmias [8]. 

In the current study, AMI seems to occur more often at midnight hours, a result that is in discordance with several previous studies that investigated the circadian distribution of AMI and demonstrated an onset of the event particularly in the morning [19,20,21]. This morning peak is reasonable and could be explained mainly by the morning increase of sympathetic activity, plasma cortisol and renin levels, BP, and HR. The increase of coronary tone, platelet aggregability, and decrease of fibrinolytic activity has also been noticed. All those factors may contribute to the unstable atherosclerotic plaque rupture and the development of AMI, especially in the morning, as shown in multiple studies [22,23,24,25]. 

In contrast, the present study highlighted an evening and a midnight peak of AMI, while a relation for AMI occurrence was also noticed for ages 40 to 65 years. These results could be explained by social and economic factors and by the fact that people at this age are mainly workers experiencing psychological, mental, and overtime working stress that may lead to a potentially acute event at the end of the day [26]. Moreover, in Greece, workers have a long lunch break, and they usually take a nap before restarting their work later in the afternoon, in contrast to Northern countries where people work continuously from the morning until evening. Therefore, the evening peak of AMI (21:00–23:59) could be related to this mid-day siesta, a characteristic of the Greek working society [27]. That is a common cultural habit in Mediterranean countries, mainly due to environmental factors. Meal habits in Mediterranean cultures might also contribute to the development of AMI or AF in the evening hours [28,29]. The whole family gathers together during dinner and usually enjoys the heaviest meal of the day [10,28,29,30,31]. The ingestion of a heavy meal with the possible consumption of large amounts of alcohol, a period of relative inactivity, and possible smoking may activate thrombotic events [29,31,32].

Another possible explanation for the alteration of the characteristic day-night pattern may be the prevalence of DM in our study. It is widely known that diabetic neuropathy, one of the most common complications of DM, may manifest for the development of “silent” myocardial infarction [33]. Patients may ignore the atypical symptoms in the morning hours and visit the emergency department at night when they are finally relaxing, and complications of MI may occur. Similar findings have been reported in other studies [16]. Hjalmarson et al. demonstrated the alteration of the characteristic morning pattern in the onset of AMI in patients with DM [34]. ANS plays an important role in the circadian pattern of CVE and suggests that in diabetic patients, abnormalities in the circadian rhythm of autonomic tone may be responsible for the altered onset of CVE [35,36]. AN is common in DM, affecting 8% of patients with recently diagnosed type 1 DM [37]. The risk of parasympathetic neuropathy increased significantly after 5 years, and the prevalence of combined AN reaches approximately 65% after a period of 10 years in patients with type 2 DM (T2DM) [38]. Diabetic AN is associated with a marked diminution of parasympathetic activation during sleep [39], and the rhythm of sympathovagal balance is significantly attenuated in patients with DM compared to those without DM [40].

In accordance with our results, a nighttime peak was also reported in patients younger than 65 who were smokers and in women with AMI with non-obstructive coronary arteries in the ACTION Registry-GWTG [41]. Lopez et al. reported that smoking is one of the most common risk factors in Mediterranean patients and causes a circadian periodicity of AMI between afternoon and midnight. They also showed a different circadian variation in the onset of AMI between 3 ethnic groups. In Mediterranean Caucasian patients, an increased number of events was noticed between midday and midnight compared to British and Indo-Asians, where the highest number of AMI onset was observed between midnight and midday [10,42].

Another finding in our study is a circadian pattern for AF initiation, mostly at noon (12:00–14:59). Intense anti-diabetic treatment, glycemic fluctuations, and different phases of insulin secretion and resistance may contribute to that pattern. Regarding our findings, by taking into consideration that lunch is the most important meal of the day, a more intensive insulin or anti-diabetic drug administration might lead to glycemic fluctuations and/or hypoglycemia. In this setting, hypoglycemia might be exacerbated by the fact that diabetic patients probably consume smaller amounts of calories during lunch, despite receiving the prescribed insulin or medication doses.

It is well established that in diabetic patients, hyperglycemia is a key mediator of atrial remodeling and initiation of AF. Electrical, electromechanical and structural remodeling, oxidative stress, connexin remodeling, autonomic remodeling and glycemic fluctuations have been implicated in the development of AF [43,44]. However, intensive glycemic control has not been found to reduce the incidence of AF, compared with standard glycemic control [45]. This finding may be related to increased episodes of severe hypoglycemia, which has been associated with sympathetic activation, shortening of the refractory period, and increased risk for AF [46,47]. Recent studies suggest that glycemic fluctuations, rather than hypoglycemia alone, contribute to the development of AF [48,49]. In a streptozotocin-induced diabetic rat model, Saito et al. demonstrated that glucose fluctuations increase the incidence of AF by promoting cardiac fibrosis [49]. Increased reactive oxygen species (ROS) levels caused by the up-regulation of thioredoxin-interacting protein (Txnip) and NADPH oxidase expression could be the underlying mechanism of glucose fluctuations-induced fibrosis [49]. In patients with DM, glucose fluctuations were more strongly correlated with increased oxidative stress than chronic hyperglycemia, suggesting that wide variations in glucose levels may be a more important risk factor for AF [50].

Another possible explanation for our results regarding the circadian pattern for AF initiation, could be the fact that in diabetic patients, insulin resistance is elevated in the morning and evening hours, but diminished at noon. Therefore, hypoglycemia may occur more often after lunch [51]. It is also well-known that insulin secretion follows two phases after meals: a rapid first and an extended second phase [52]. In patients with T2DM, there is reduced insulin secretion during the first phase and extended insulin secretion during the second phase, a fact that may also lead to hypoglycemia.

## 5. Limitations

The main limitation of our study is the relatively small sample of the patients. Drugs reducing BP and lowering HR, lipid levels, and platelet aggregation might also reduce the plaque vulnerability to external factors. A characteristic example is b-blockers that have been showed to dull the circadian pattern in AMI by reducing BP and HR and increasing coronary blood flow by prolonging the diastolic phase [53]. Our study did not take into consideration the medication of patients. Patients with AMI and AF without a history of DM or with a history of DM less than 5 years were excluded from our study. Moreover, the patients with AMI or AF that died at the emergency department or outside the hospital were not included in the study. We relied on medical reports, patient self-reported history, and drug history to identify DM before hospitalization, and on glucose calculation during hospitalization.

## 6. Conclusions

Circadian pattern plays an important role in CVE and arrhythmias. Our study revealed that AMI and AF follow a specific circadian pattern in diabetic patients in a Mediterranean country. AMI seems to occur more often at midnight hours, a result that is in discordance with previous studies reporting that the majority of MI occurs mainly in the morning hours. Autonomic alterations, as well as working habits and social reasons, might contribute to this phenomenon. AF also follows a circadian pattern and is initiated mostly at noon. Glycemic fluctuations, intense anti-diabetic treatment before lunch, and patterns of insulin secretion and resistance may play a role. More studies are needed to elucidate the circadian pattern of AMI and AF in diabetic patients and investigate several other related parameters to contribute to the development of new therapeutic approaches in this setting.

## Figures and Tables

**Figure 1 medicina-57-00041-f001:**
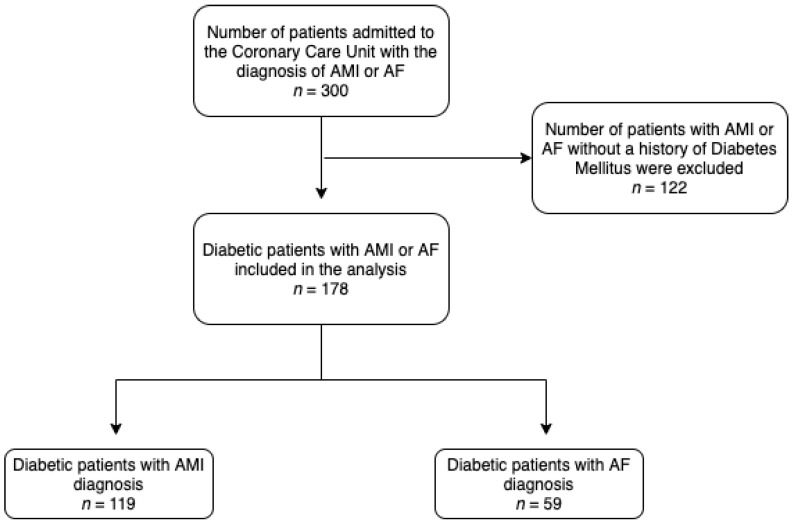
Retrospective design and flow chart of the study. AMI: acute myocardial infarction; AF: Atrial Fibrillation.

**Figure 2 medicina-57-00041-f002:**
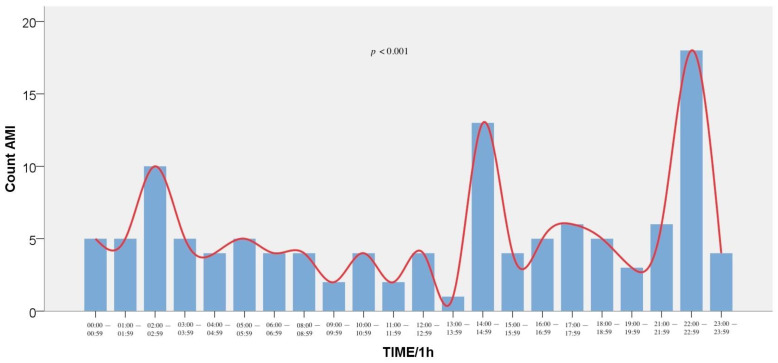
Frequency of Acute Myocardial Infarction incidents by hour of symptom onset (1-h intervals).

**Figure 3 medicina-57-00041-f003:**
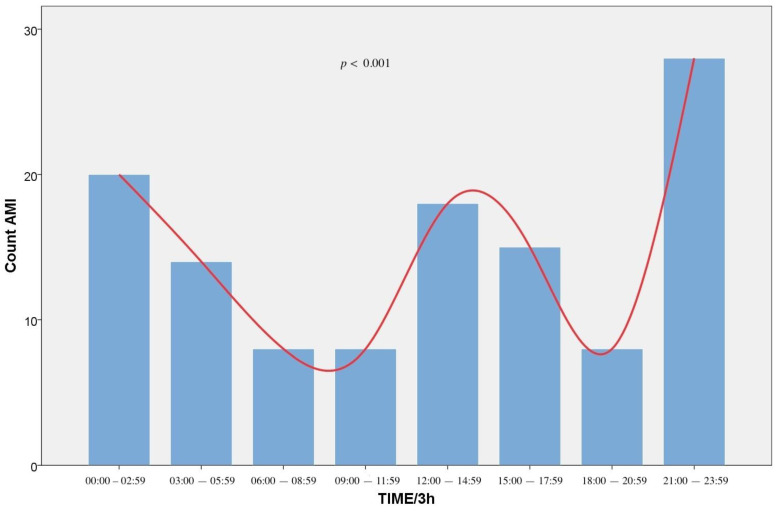
Frequency of Acute Myocardial Infarction incidents (AMI) by hour of symptom onset (3-h intervals). The plot indicates that AMIs were more frequent during midnight hours.

**Figure 4 medicina-57-00041-f004:**
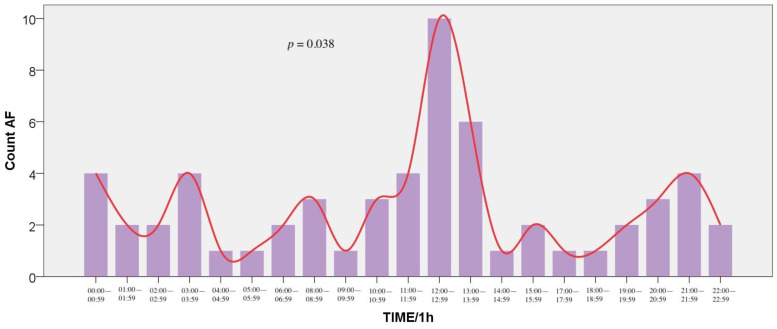
Frequency of Atrial Fibrillation incidents by hour of symptom onset (1-h intervals).

**Figure 5 medicina-57-00041-f005:**
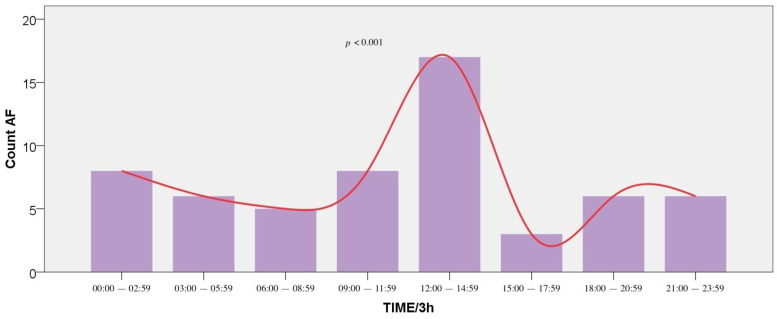
Frequency of Atrial Fibrillation (AF) incidents by hour of symptom onset (3-h intervals). The plot indicates that AF events were more frequent at noon.

**Figure 6 medicina-57-00041-f006:**
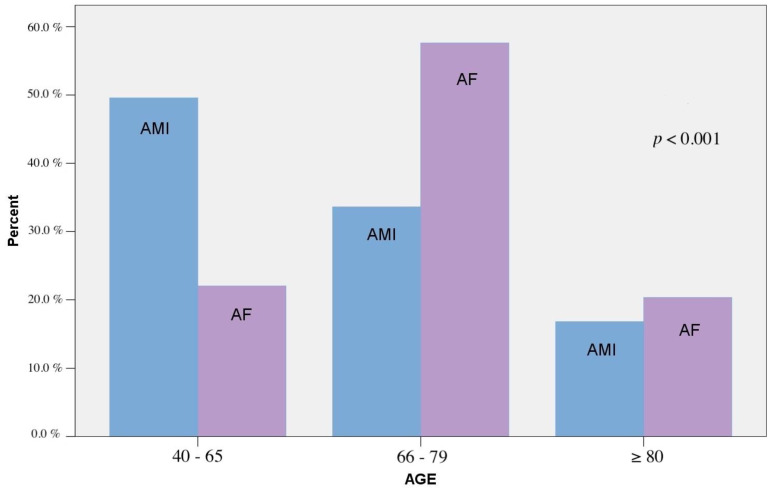
Age distribution of Acute Myocardial Infarction and Atrial fibrillation. Age divided in 3 age groups for calculation purposes.

**Table 1 medicina-57-00041-t001:** Patients’ demographic and clinical characteristics.

Demographic Characteristics	*n*/Mean	%/SD
Number of patients	178	
Age	67.7	12.0
40–65	72	40.4
66–79	74	41.6
≥80	32	18.0
Male	123	69.1
Female	55	30.9
AMI	119	66.9
AF	59	33.1
Medical history		
Hypertension	123	69.1
Smokers	131	73.6
Dyslipidaemia	55	30.9

AMI: acute myocardial infarction; AF: Atrial Fibrillation.

**Table 2 medicina-57-00041-t002:** Patients’ characteristics classified by age group.

Characteristics	Age Group			
	Overall	40–65	66–79	80–89
Patients enrolled	178	72	74	32
*Gender, n (%)*				
Male	123 (69.1)	58 (80.6)	46 (62.1)	19 (59.4)
Female	55 (30.9)	14 (19.4)	28 (37.8)	13 (40.6)
*Event, n (%)*				
AMI	119 (66.9)	59 (81.9)	40 (54.1)	20 (62.5)
AF	59 (33.1)	13 (18.1)	34 (45.9)	12 (37.5)
*Medical History, n (%)*				
Hypertension	123 (69.1)	45 (62.5)	51 (68.9)	27 (84.4)
Smoking	131 (73.6)	55 (76.4)	54 (73)	22 (68.8)
Dyslipidemia	55 (30.9)	26 (36.1)	22 (29.7)	7 (21.9)

## Data Availability

The data presented in this study are available on request from the corresponding author. The data are not publicly available due to privacy restrictions.

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
