# Peer review of "Circadian Pattern of Acute Myocardial Infarction and Atrial Fibrillation in a Mediterranean Country: A study in Diabetic Patients"

_medicina, 2021, doi:10.3390/medicina57010041_

Round 1

Reviewer 1 Report

Dear Authors,

I have read the current revised version of your manuscript “Circadian Pattern of Acute Myocardial Infarction and Atrial Fibrillation in a Mediterranean Country: a study in Diabetic patients” with great attention.

All the raised questions were fully answered and followed by the appropriate amendments to the manuscript - therefore my recommendation to the Editor is to consider the acceptance of the manuscript in the current form.

It was a pleasure to review your paper and let me personally congratulate you on your work!

Author Response

Dear Editorial Board,

Dear Reviewers,

With Regard to Reviewer #1:

We were very pleased to receive the evaluation of our manuscript and we would like to express our sincere gratitude to the editor and the reviewers for their insightful comments. In particular, we appreciate your feedback for accepting the paper at its current revised-resubmitted form. We are glad to know that all your raised questions and appropriate amendments were fully answered.

We declare there is no conflict of interest in our submission and all authors have

approved the final version of the manuscript.

Overall, in our revised work we incorporated modifications adhering to the

comments / suggestions / requests by the Reviewers. We hope that you will find them satisfactory and our improved manuscript will be deemed suitable for publication Medicina Journal.

We look forward to hearing from you.

Faithfully yours,

Christos Savopoulos on behalf of the authors

First Propedeutic Department of Internal Medicine

AHEPA University Hospital

Medical School, Aristotle University of Thessaloniki

St. Kiriakidi 1, 54636 Thessaloniki, Greece

Tel: +302310994779

Reviewer 2 Report

Reviewer comments and suggestions

The current research paper tried to study whether acute myocardial infarction (AMI) and atrial fibrillation (AF) monitored a circadian pattern in diabetic patients in a Mediterranean country. The current study was a retrospective study that included 178 diabetic patients (mean age: 67,7) with AMI or AF who were admitted to the coronary care unit of the hospital. The study reports found that AMI seems to occur more often in the midnight hours (21:00-23:59) (p<0.001). Regarding age distribution, patients between 40 and 65 years were more likely to have risk for AMI compared to other age groups (p<0.001). On the other hand, AF in diabetic patients occurs more frequently at noon (12:00-14:59) (p=0,019). The study inference was that diabetic patients with AMI and AF seem to follow a specific circadian pattern in a Mediterranean country

Basically, the representation of the paper was not good. Some of the descriptive tables classified based on the age group should be present in the main MS and a flow chart should be present in the paper to clarify exactly what number of the patients were recruited and how any of them were out from the study. In an Epidemiological study, it is very important to understand the clear cut study design of the populations. So please explore more material and method section in the paper. Besides the above, I am suggesting some minor comments to be incorporated in the revised version of the MS.

Below are the comments that need to update in the revised version of the manuscript.

  1. Line 43-44, no need to add this, the author should present the results only when they have demonstrated in the paper, I mean the data analyzed by the authors.

Better to highlight CP in the conclusion line. What type of circadian pattern they found in the patients, need to present here? Not only discussing like a specific circadian

Line 65, The author needs to define it before going to discuss the things

Line 74, reference 10 should be explored in detail

Line 80, better not be abbreviated first time used (DM)

  1. Line 88, Cardiovascular Events (CVE) C ANDE should be a small letter
  2. Line 91, please mention the exact hospital name, location, and also there should be ethical approval for conducting the study.
  3. Line 100, Grammatical mistake
  4. Line 127, Please draw a flow chart for the study design
  5. Please describe 3-hour equal intervals, line 150
  6. Line 154, is there was any specific reason for this
  7. Line 225, Here the author has to write the outcome of the study
  8. Line 230, Why this study was discordant with other previous studies, please explain with some reason for this
  9. Line 234, did the author check the parameters
  10. Line 247-248, specific reason
  11. Line 250-251, it is important to correlate your study with other relevent published studies
  12. Line 253-255, need a reference
  13. Line 262-263, as study related to type 2, so is it important to cite type 1 diabetes here
  14. Line 273-274, not understandable please simply the sentences
  15. Line 319, it should be some bias, please name what kind of bias
  16. Line 331, did you check insulin resistance
  17. Line 333-334, better to study other related parameters of diabetes.
  18. Please check the reference number, 26,27, 48 and 50

Author Response

Dear Editorial Board,

Dear Reviewers,

With Regard to Reviewer #2:

We were very pleased to receive the evaluation of our manuscript and we would like to thank you and the reviewers for your insightful comments. We would like to express our sincere gratitude for providing us with valuable feedback and suggestions. We have addressed all the concerns raised in a revised version of the manuscript as requested. All changes in the manuscript are highlighted in yellow format (resubmitted file), due to the fact that it was accepted in the resubmitted form by reviewer #1 and in Green Format (revised file), regarding to your suggestions in the respective file. Please find below our response, point-by-point, to the comments provided by the Reviewers.

The current research paper tried to study whether acute myocardial infarction (AMI) and atrial fibrillation (AF) monitored a circadian pattern in diabetic patients in a Mediterranean country. The current study was a retrospective study that included 178 diabetic patients (mean age: 67,7) with AMI or AF who were admitted to the coronary care unit of the hospital. The study reports found that AMI seems to occur more often in the midnight hours (21:00-23:59) (p<0.001). Regarding age distribution, patients between 40 and 65 years were more likely to have risk for AMI compared to other age groups (p<0.001). On the other hand, AF in diabetic patients occurs more frequently at noon (12:00-14:59) (p=0,019). The study inference was that diabetic patients with AMI and AF seem to follow a specific circadian pattern in a Mediterranean country.

Basically, the representation of the paper was not good. Some of the descriptive tables classified based on the age group should be present in the main MS and a flow chart should be present in the paper to clarify exactly what number of the patients were recruited and how any of them were out from the study. In an Epidemiological study, it is very important to understand the clear cut study design of the populations. So please explore more material and method section in the paper. Besides the above, I am suggesting some minor comments to be incorporated in the revised version of the MS.

-Thank you very much for your kind comments. We improved the representation of the paper by adding a flow chart (Figure 1) and descriptive tables (Table 2) based on the age group as requested and explored more material and method section. We also tried to cover all the minor comments and incorporate them in the revised version of the MS. Below are the comments that were updated in the revised version of the manuscript. We would like to mention that due to the fact that we included more sentences and references, the line numbering of the revised manuscript was modified. All changes in the manuscript are highlighted in yellow format (resubmitted file) and in Green Format (revised file as requested) in the respective file. Please find below our response, point-by-point, to the comments provided by the Reviewer#2.

1.Line 43-44, no need to add this, the author should present the results only when they have demonstrated in the paper, I mean the data analyzed by the authors.

-According to your recommendation the phrase “Glycemic fluctuations, intense anti-diabetic treatment before lunch, patterns of insulin secretion and resistance may explain this pattern” (Line 43-44) was removed from the results section and added into the conclusion line.

Better to highlight CP in the conclusion line. What type of circadian pattern they found in the patients, need to present here? Not only discussing like a specific circadian

-According to your recommendation the phrase “with AMI occurring most often at midnight hours and AF mostly at noon” was added in the conclusion section to specify the type of the circadian pattern that was followed.

Line 65, The author needs to define it before going to discuss the things

-The period “Circadian rhythm refers to a biological process that modulates the internal 24-hour clock of the organism and regulates the cycles of alertness and sleepiness” was added with the corresponding citation to define the circadian rhythm entity.

Line 74, reference 10 should be explored in detail

-According to your recommendation the citation number [10] was added. We also added a phrase to explain the “siesta” effect as you proposed “More specifically, increased BP and HR was reported after the evening-nap in a similar manner after waking up in the morning.”

Line 80, better not be abbreviated first time used (DM)

-The abbreviation DM was corrected and converted to Diabetes Mellitus (DM).

6.Line 88, Cardiovascular Events (CVE) C ANDE should be a small letter

-“Cardiovascular Events” was converted to “cardiovascular events” with small starting letters

7.Line 91, please mention the exact hospital name, location, and also there should be ethical approval for conducting the study.

-The study was conducted at Serres General Hospital, Greece with the contribution of First Propedeutic Department of Internal Medicine, AHEPA university hospital,Thessaloniki, Greece. In our country, in order to conduct a non-invasive study, all the required documents should be signed by a scientific council, taking place once per month. All the required documents regarding our study were signed by the scientific council (Protocol Number: 20240) of our hospital and the director of the CCU and Cardiology Department. All the patients were informed about their participation in the study.

8.Line 100, Grammatical mistake

-The phrase was corrected into “A convenience sample of 178 patients with DM admitted to the coronary care unit (CCU) were the study’s participants”

9.Line 127, Please draw a flow chart for the study design

-We drew a flow chart (Figure 1) and presented it into the MS file

10.Please describe 3-hour equal intervals, line 150

-The 3-hour equal intervals are described in Figure 3. We also added them in brackets in line 150 according to your recommendation; “00:00-02:59, 03:00-05:59, 06:00-08:59, 09:00-11:59, 12:00-14:59, 15:00-17:59, 18:00-20:59,21:00-23:59”.

11.Line 154, is there was any specific reason for this

-We mainly explain the fact that younger patients experienced an AMI more frequently than the older ones in the discussion section in the following paragraph “These results could be explained by social and economic factors and by the fact that people at this age are mainly workers experiencing psychological, mental, and overtime working stress that may lead to a potential acute event at the end of the day. Moreover, in Greece, workers have a long lunch break, and they usually take a nap before restarting their work later in the afternoon in contrast to Northern countries where people work continuously from the morning until evening. Therefore, the evening peak of AMI (21:00-23:59) could be related to this mid-day siesta, a characteristic of the Greek working society”.

-More particularly, younger patients are workers experiencing psychological, mental and overtime working stress that may function as a triggering factor for an acute event at the end of the day. The nighttime peak of AMI comes also in accordance with ACTION Registry-GWTG.

12.Line 225, Here the author has to write the outcome of the study

-The main outcome of the study was added “Our study presents a circadian pattern of AMI mainly at midnight hours (21:00-23:59) and a higher incidence of them in patients in 40 to 65 years. Regarding the AF initiation, the events occurred more frequently at noon (12:00-14:59).” according to your recommendation.

13.Line 230, Why this study was discordant with other previous studies, please explain with some reason for this

-We tried to explain the reasons of discordance further in the discussion. The population included in our study were patients of a Mediterranean country with different cultural habits and environmental factors. The phenomenon of Greek “siesta” is also highlighted, that seems to contribute significantly in the circadian clock of each patient. Indeed, a raised heart rate and blood pressure have been demonstrated immediately after the siesta, in a very similar fashion that happens after waking up in the morning, leading to an increase in cardiovascular events (An explanation of the siesta effect was added in the introduction section). The prevalence of Diabetes mellitus in our study seems to play a role as well. That is also explained furtherly in the discussion section.

14.Line 234, did the author check the parameters

-All the researched parameters were mentioned in the study. This is the reason that we mention these parameters as a possible explanation and not as a definite explanation.

15.Line 247-248, specific reason

-We would like to clarify that the reason is explained in line 250-251 “The ingestion of a heavy meal with the possible consumption of large amounts of alcohol, a period of relative inactivity, and possible smoking may activate thrombotic events”. More specifically Kolettis et al. reported that the consumption of a large meal in the evening leads to a period of physical inactivity as it is mentioned in our discussion. Ingestion of alcohol may act synergistically. We also added another reference to correlate our study with other relevant studies (Ref. 32).

16.Line 250-251, it is important to correlate your study with other relevent published studies

-Another reference was added according to your recommendation (number 32); “Peters RW, Zoble RG, Liebson PR, et al. Identification of a secondary peak in myocardial infarction onset 11 to 12 h after awakening: the Cardiac Arrhythmia Suppression Trial (CAST) experience. J Am Coll Cardiol 1993;22:998–1003.” In total 3 relevant references were correlated with line 250-251

17.Line 253-255, need a reference

-We added a reference at the line 253-255 (number 33) “O'Sullivan JJ, Conroy RM, MacDonald K, et al. Silent ischaemia in diabetic men with autonomic neuropathy. Br Heart J. 1991;66(4):313-315. doi:10.1136/hrt.66.4.313”. Thus, the number of all references was changed.

18.Line 262-263, as study related to type 2, so is it important to cite type 1 diabetes here

-The citation is related to both type 1 and type 2 diabetes mellitus. More particularly, “Criteria for entry into the study included the presence of either Type 1 (insulin-dependent) diabetes defined by serum C-peptide < 0.10 nmol I-’ after stimulation with 1 mg glucagon i.v. and/or history of hypergIycaemia/ketonuria with requirement for insulin from the time of diagnosis or Type 2 (non-insulin-dependent) diabetes classified according to the proposals of the National Diabetes Data Group,20 and duration of diabetes ≥ 1 year”.

-However, if your think that we should modify the citation, “Ziegler, D., et al. The natural history of somatosensory and autonomic nerve dysfunction in relation to glycaemic control during the first 5 years after diagnosis of Type 1 (insulin-dependent) diabetes mellitus. Diabetologia, 1991;34(11), 822–829. doi:10.1007/bf00408358” can be added.

19.Line 273-274, not understandable please simply the sentences

-We simplified the sentence and converted it to “In Mediterranean Caucasian patients, an increased number of events was noticed between mid-day and midnight compared to British and Indo-Asians, where the highest number of AMI onset was observed between midnight and mid-day”.

20.Line 319, it should be some bias, please name what kind of bias

-We added the phrase “Our study was subject to recall bias: we relied on medical reports, patient self-reports and medication history to identify DM before hospitalization, and on glucose calculation during hospitalization.”

21.Line 331, did you check insulin resistance

-All the parameters that were investigated were mentioned in the manuscript. However, we will keep in mind the idea of organizing a protocol about the circadian pattern of Atrial Fibrillation in Diabetic patients (including insulin resistance for each patient).

22.Line 333-334, better to study other related parameters of diabetes.

-The phrase “and investigate several other related parameters” was added to the conclusion section according to your recommendation

23.Please check the reference number, 26,27, 48 and 50

-Thank you very much for clarifying which references were flawed. After your contribution we corrected the reference 26,27,48,50 (We corrected the references that you mentioned). However, the numbering of references was changed due to the incorporation of additional references.

We declare there is no conflict of interest in our submission and all authors have

approved the final version of the manuscript.

Overall, in our revised work we incorporated modifications adhering to the

comments / suggestions / requests by the Reviewers. We hope that you will find them

satisfactory and our improved manuscript will be deemed suitable for publication Medicina Journal.

We look forward to hearing from you.

Faithfully yours,

Christos Savopoulos on behalf of the authors

First Propedeutic Department of Internal Medicine

AHEPA University Hospital

Medical School, Aristotle University of Thessaloniki

St. Kiriakidi 1, 54636 Thessaloniki, Greece

Tel: +302310994779

This manuscript is a resubmission of an earlier submission. The following is a list of the peer review reports and author responses from that submission.

Round 1

Reviewer 1 Report

Dear Authors,

I have read submitted manuscript “Circadian Pattern of Acute Myocardial Infarction and Atrial Fibrillation in a Mediterranean Country: a study in Diabetic patients” with great attention. This is an interesting retrospective single-center analysis regarding the above subject, the submitted manuscript is well written and it was a pleasure to read, however it will still benefit with some minor corrections:

  1. In Introduction section (page 3, lines 61-63) it is stated: “A different circadian periodicity has been observed...” Different from what? It is not specified in this text passage (however discussed later) and can be confusing to the reader. This should be corrected.

  1. All abbreviations should be explained at first mention in the text (page 4, line 70) “Specifically, BMAL1 binds and initiates...”.

  1. In the Methods section (page 6, lines 112-114) the age groups are not precisely specified „...the first group included patients from 40 to 65 years, the second group from 65 to 79 years...” With such description, it is unclear where a 65-year-old patient should belong? Moreover, the correct age group description is provided further in the text (page 6, lines 126-127). Perhaps this is an unnecessary duplication and you can just delete one of them?

  1. In the Limitations section (page 16, lines 300-302) you declared the patients that have died during hospitalization were not included in the study. If so, this should be also listed in the Method section as the exclusion criterion, and the rationale discussed. To be precise, the title perhaps should be modified to “Circadian Pattern of Non-Fatal Acute Myocardial Infarction...”?

This minor comments do not affect the great value of your work, which I do appreciate.

Author Response

Dear editorial office of Medicina Journal, dear editor-in-chief, dear reviewer 1

Thank you very much for your valuable contribution regarding our manuscript review. After reading very carefully your comments, our response is as follows:

  1. In the Introduction section (page 3, lines 61-63), we modified the phrase “A different circadian periodicity has been observed” with the phrase “A specific circadian periodicity has been observed”, as you pointed out because there is no need for a comparison to be made.

  1. After your suggestion we explained the BMAL1 further into Brain and muscle Arnt-like protein 1 and the Scn5a gene into Sodium voltage-gated channel alpha subunit 5.

  1. In the Methods section (page 6, lines 112-114) we corrected the sentence “the first group included patients from 40 to 65 years, the second group from 65 to 79 years...” into “the first group included patients from 40 to 65 years, the second group from 66 to 79 years...”, as you clearly observed. We suppose that it is necessary to mention the age groups analytically in the methods section and briefly further in the text (page 6, lines 126-127).

  1. We included in the Methods section (data collection subsection) the exclusion criterion and added a sentence explaining the rationale behind this decision “Patients with AMI or AF that died at the emergency department or during their hospitalization were not included in the study due to the fact that we could not obtain the medical records of the patients. Our hospital’s system does not give us the possibility of recording and creating a medical record for the patients that either visited the ED and were not hospitalized or died”. The number of patients who died during their hospitalization was negligible (1 patient), so we decided to exclude him as well, as this does not affect the statistical significance of the final result and obtaining medical records from the hospitalized patients who died during their hospitalization remains extremely difficult. Furthermore, If you consider the title change obligatory, we could modify it.

We improved the discussion part by majorly revising it according to the comments of the other reviewers and the editor. In addition, we proceeded to English style editing, and we corrected some minor mistakes.

Reviewer 2 Report

the paper is well written in a simple and clear way. Topic is not so original and study has important limitations such as number of enrolled patients. Discussion is interesting but your conclusions are assumptions that your data can not confirm.

Author Response

Dear editorial office of Medicina Journal, dear editor-in-chief, dear reviewer 2

Thank you very much for your valuable contribution regarding our manuscript review. After reading very carefully your comments, our response is as follows:

Our study is a single-center observational study, and in order to highlight our conclusions, we have improved our discussion and revised the text of our manuscript according to your and also to the editor’s comments.

The conclusion of our research is now clearly analyzed in the discussion section by several results that concern the interpretation of our data based on the characteristics of our population, a population of a small Mediterranean country with great homogeneity but with several peculiarities (working habits, social and family reasons, climate). The explanation is based on the pathophysiology mechanisms explained and concerns our conclusion one by one with all the sources cited.

We also furtherly improved the methods section according to the comments of reviewer 1 and proceeded to English style editing, correcting some minor mistakes. We uploaded the revised manuscript in a track changes format to demonstrate the changes that have been made.